# The case for decreased surgeon-reported complications due to surgical volume and fellowship status in the treatment of geriatric hip fracture: An analysis of the ABOS database

Taylor D. Ottesen[1,2], Michael R. Mercier[1], Jordan Brand[1], Michael Amick[1], Jonathan N. Grauer[1]*, Lee E. Rubin[1]

**1** Department of Orthopaedics and Rehabilitation, Yale School of Medicine, New Haven, Connecticut, United States of America, **2** Harvard Combined Orthopaedic Residency Program, Boston, Massachusetts, United States of America

* Jonathan.grauer@yale.edu

## Abstract

### Introduction

American orthopaedists are increasingly seeking fellowship sub-specialization. One proposed benefit of fellowship training is decrease in complications, however, few studies have investigated the rates of medical and surgical complications for hip fracture patients between orthopedists from different fellowship backgrounds. This study aims to investigate the effect of fellowship training and case volume on medical and surgical outcomes of patient following hip fracture surgical intervention.

### Methods

1999–2016 American Board of Orthopedic Surgery (ABOS) Part II Examination Case List data were used to assess patients treated by trauma or adult reconstruction fellowship-trained orthopedists versus all-other orthopaedists. Rates of surgeon-reported medical and surgical adverse events were compared between the three surgeon cohorts. Using binary multivariate logistic regression to control of demographic factors, independent factors were evaluated for their effect on surgical complications.

### Results

Data from 73,427 patients were assessed. An increasing number of hip fractures are being treated by trauma fellowship trained surgeons (9.43% in 1999–2004 to 60.92% in 2011–2016). In multivariate analysis, there was no significant difference in type of fellowship, however, surgeons with increased case volume saw significantly decreased odds of complications (16–30 cases: OR = 0.91; 95% CI: 0.85–0.97; p = 0.003; 31+ cases: OR = 0.68; 95% CI: 0.61–0.76; p<0.001). Femoral neck hip fractures were associated with increased odds of surgical complications.

**Data Availability Statement:** Data is only made available by application to the ABOS Research Committee. A fee is typically associated with

access to this data. If approved, data is therefore made available and provisioned by Mona Sanlei. The authors did not have any special privileges in access the data from the ABOS that other researchers would not have. Pending approval, any research would be allowed access to the data we used to conduct the current study.

**Funding:** The author(s) received no specific funding for this work.

**Competing interests:** The authors have declared that no competing interests exist.

## Discussion

Despite minor differences in incidence of surgical complications between different fellowship trained orthopaedists, there is no major difference in overall risk of surgical complications for hip fracture patients based on fellowship status of early orthopaedic surgeons. However, case volume does significantly decrease the risk of surgical complications among these patients and may stand as a proxy for fellowship training. Fellows required to take hip fracture call as part of their training regardless of fellowship status exhibited decreased complication risk for hip fracture patients, thus highlighting the importance of additional training.

## Introduction

Hip fractures are a major source of morbidity and mortality affecting an estimated 340,000 persons annually and projected to grow to over 600,000 annually by 2040 in the United States [1, 2]. Usually arising after minor trauma, typically a fall from standing height [3], fragility hip fractures among elderly patients have been shown to seriously affect physical and mental functioning and severely impact their health status and health-related quality of life [3]. As such, proper treatment of hip fractures is essential to minimize complications, decrease length of stay, and maximize recovery [4].

Treatment for hip fractures varies from hospital to hospital, and often includes treatment by general orthopaedic surgeons, orthopaedic trauma surgeons, or other fellowship trained orthopaedic surgeons [5]. Research continues to evaluate the respective care provided as a result of sub-specialization training.

Over the past decade, orthopaedic surgery has become increasingly specialized for a variety of reasons: providing extra training in specific areas of interest, improving clinical expertise, and optimizing chances to ensure employment [6]. In 2012, it was estimated that 87.4% of orthopaedic graduates pursue additional training in a subspecialized fellowship [6, 7]. While orthopaedic residency exposes residents to generalized practices, fellowship can ensure subspecialized orthopaedic operative competency and increased caseloads to prepare surgeons for the evolving nature of the field [7, 8].

Increased surgical volume has been extensively studied across a wide variety of procedures to investigate the link between experience and surgeon skill and patient complications [9–15]. In the case of hip fractures specifically, greater surgical volume has shown mixed findings in relation to surgical complications [16–21]. While some studies show surgical volume to be associated with decreased mortality [17, 18], others have shown no such associations [20, 21]. Similarly, the treatment of hip fractures by fellowship trained surgeons and general orthopaedic surgeons has also shown mixed results. Treatment of intertrochanteric fractures by trauma and non-trauma orthopaedic surgeons showed no difference in complications [5], however another study comparing fellowship trained surgeons (trauma or arthroplasty) and general orthopaedic surgeons did show differences in complication rates and 1-year mortality following treatment via hemiarthroplasty [22]. While these findings debate the importance of surgical volume and fellowship training, the implications of the effect of surgical volume on early practice physicians requires further attention. Additionally, the act of fellowship training itself on post-operative complications remains understudied.

Due to this dearth of information, the current study aimed to identify the pertinent factors in the occurrence of surgeon-reported complications of hip fractures in the training, fellowship specialization type, and experience of early practice orthopaedic surgeons and trend those over an extended period of time. Utilization of data from 1999 through 2016 American Board of Orthopaedic Surgery (ABOS) database enables a large, longitudinal review of operative treatment and subsequent complications for hip fracture patients treated by different fellowship-trained surgeons. Such analysis will provide insight and direction for orthopaedic graduates and hospital administrators in an ever increasingly subspecialized field [23]. Continuous evaluation and assessment of orthopaedic fellowship sub-specialization aims to decrease complication rates for patients and ensure better training and preparation for orthopaedic residency graduates as they transition into their careers.

## Material and methods

### Background

In the United States, the American Board of Orthopaedic Surgery (ABOS) awards Board Eligibility and then Board Certification to physicians completing all necessary requirements after finishing a five-year accredited orthopaedic residency program in either the US or Canada. A major component of this certification is the successful completion of a two-part examination–Part I is given via a computer-based multiple-choice examination and Part II is given as an oral examination [24]. After finishing the computer-based Part I, applicants become ABOS Board Eligible and are required to submit a comprehensive list of all surgical procedures they perform over a period of six months during a practice period of 20 months at one practice location. They must submit a minimum of 35 cases for review [25–27]. The list of procedures is logged by the applicant and includes a wide range of data including the applicant's fellowship status, patient demographics (e.g. patient age and gender) and information about the procedure (e.g. Current Procedural Terminology (CPT) codes, and anesthetic, medical, and surgical complications). Once this data is received and reviewed, applicants can be approved to take the ABOS Part II Oral Examination. All of the patient information submitted as a part of this process is then de-identified and made available for research purposes after application and approval through the ABOS Research Committee.

### Data

For the current study, we investigated the 1999–2016 databases for geriatric patients (defined as age > 65) who were treated for hip fracture by an ABOS candidate during this period. Patients were identified using the following CPT codes and author assigned categories: 27235 (simple percutaneous pinning hip fracture), 27236 (femoral neck hip fracture), or 27244/27245 (intertrochanteric hip fracture). Patients were then grouped based on fellowship status of the treating physician (trauma, adult reconstruction, or other). Physicians in the "other" fellowship category included any general orthopedist applicant (did not complete a fellowship) as well as candidates who had completed a fellowship in a field other than trauma or adult reconstruction. If a candidate had completed a fellowship in both trauma and adult reconstruction, their cases were excluded to ensure clarity of findings between the fellowships. Patients were also grouped by number of cases completed by the candidate during their collection period and put into the following groupings: 0–15 cases, 16–30 cases, and 31+ cases. Lastly, candidates were broken up by region of the United States based on the pre-defined regions as defined by the ABOS Research Committee (Fig 1).

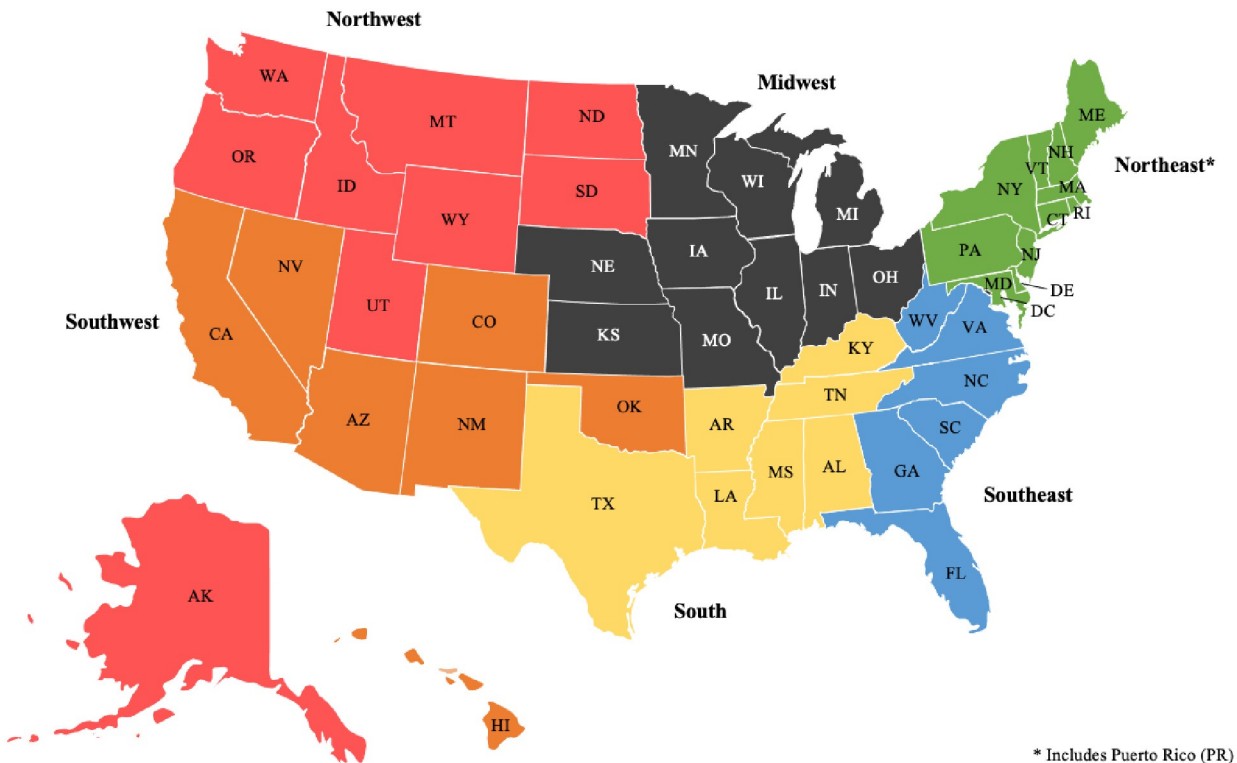

**Fig 1. Pre-defined regions of the United States for ABOS candidates.**

## Analysis

All available demographics for each patient was extracted including patient age and sex, geographic region, the year of the procedure, and procedural codes. As coding of complications has changed in the ABOS database in recent years, the current study identified surgeon-reported medical complications and surgeon-reported surgical complications that were collected across all years of data collection to ensure consistency of findings. Collected data that was not present across all years was excluded to ensure consistent complication rate data and comparisons. The medical complications recorded across all years and subsequently extracted for analysis were: death, myocardial infarction, stroke, renal failure, pulmonary embolism, congestive heart failure, pneumonia, and medical unspecified complications. The surgical complications recorded across all years were: bone fracture, dislocation, infection, nonunion/delayed union, skin ulcer/blister, implant failure, nerve palsy/injury, and vascular injury. Binary logistic regressions investigating likelihood of complication based on demographic covariates of patients and treating physician was then performed with patient age, patient sex, region, year of procedure, fellowship status, procedure type, and number of cases preformed.

IBM SPSS Statistics for Macintosh (version 26; IBM Corp., Armonk, NY) was used for all analysis. Chi-square tests were performed for the categorical variables between the three cohorts (both fellowship-type and case-number). Student t-test was utilized for continuous variables.

The ABOS Research Committee approved the current study and the study was classified as exempt by our institutional review board.

## Results

The current study identified 73,427 hip fracture surgeries that fit inclusion criteria. Of these, trauma surgeons performed 14.67% (10,769), adult reconstruction performed 14.64% (10,748), and other fellowship trained surgeons performed the remaining 70.70% (51,910) hip fracture procedures (Table 1). Across surgeries performed, the average ages for cases were 82.17, 82.79, and 82.64 for trauma, adult reconstruction, and other surgeons respectively (p<0.001). A statistically significant but clinically insignificant difference in percent of surgeries performed in female patients was seen between the groups (between 71.59% and 73.46%, p < 0.001). From 1999–2016, trauma and adult reconstruction surgeons increased their share of hip fracture procedures while those surgeons in other fellowships saw a decline in their total hip fracture procedures in 2011–2016. Looking at breakdown of type of procedure done, surgeons in all fellowship categories performed the greatest percentage of intertrochanteric hip fractures in relation to all other cases they performed (Table 1).

In examining the incidence of surgeon-reported medical adverse events, fellowship training was not found to have any significant overall effect on most adverse medical events. However, in the case of renal failure (p<0.001), pulmonary embolism (p = 0.008), and pneumonia (p = 0.002), trauma surgeons were more likely to have such complications following hip fracture surgeries (Table 2).

In the setting of surgeon-reported surgical adverse events on the other hand, trauma surgeons were the least likely to have any surgical complications compared to the adult reconstruction and other fellowship trained counterparts with complication rates of 5.44%, 5.95%, and 6.19% respectively (p = 0.01) (Table 3). These differences in surgical complications were

**Table 1. Demographics of hip fracture patients, organized by fellowship type of the treating physician and procedure type.**

| Type: Hip Fracture | Trauma | Adult Reconstruction | Other* | Univariate P-value |
|---|---|---|---|---|
| N = 73,427 (100%) | N = 10,769 (14.67%) | N = 10,748 (14.64%) | N = 51,910 (70.70%) | |
| **Age (SD)** | 82.17 (8.22) | 82.79 (7.83) | 82.64 (7.81) | <**0.001** |
| **Sex** | | | | |
| Male | 3,060 (28.41%) | 2,882 (26.81%) | 13,778 (26.54%) | <**0.001** |
| Female | 7709 (71.59%) | 7,866 (73.19%) | 38,132 (73.46%) | |
| **Region** | | | | |
| Northeast | 2,172 (20.17%) | 2,941 (27.36%) | 9,888 (19.05%) | <**0.001** |
| Northwest | 461 (4.28%) | 467 (4.34%) | 3,401 (6.55%) | |
| Midwest | 2,325 (21.59%) | 2,225 (20.70%) | 10,768 (20.74%) | |
| South | 1,787 (16.59%) | 1,539 (14.32%) | 9,989 (19.24%) | |
| Southeast | 2,016 (18.72%) | 1,910 (17.77%) | 8,967 (17.27%) | |
| Southwest | 1,949 (18.10%) | 1,650 (15.35%) | 8,630 (16.62%) | |
| **Year of Procedure**** | | | | |
| 1999–2004 | 1,016 (6.79%) | 1,894 (12.67%) | 12,044 (80.54%) | <**0.001** |
| 2005–2010 | 3,192 (11.13%) | 4,004 (13.96%) | 21,481 (74.91%) | |
| 2011–2016 | 6,561 (22.02%) | 4,850 (16.28%) | 18,385 (61.70%) | |
| **Procedure Type (CPT Code)** | | | | |
| Simple Percutaneous Pinning Hip Fracture (27235) | 1,125 (10.45%) | 1,079 (10.04%) | 5,821 (11.21%) | <**0.001** |
| Femoral Neck Hip Fracture (27236) | 3,612 (33.54%) | 4,028 (37.48%) | 18,356 (35.36%) | |
| Intertrochanteric Hip Fracture (27244, 27245) | 6,032 (56.01%) | 5,641 (52.48%) | 27,733 (53.43%) | |

*Other fellowship specialties include: foot and ankle, pediatrics, sports medicine, hand/upper extremity, oncology, spine, and shoulder/elbow

**Percentages were calculated horizontally to better reflect changes in each time strata

**Table 2. Incidence of medical adverse events following all hip fracture surgeries, organized by fellowship type of the treating physician.**

| Type: Hip Fracture | Trauma | Adult Reconstruction | Other | Univariate P-value |
|---|---|---|---|---|
| N = 73,427 (100%) | N = 10,769 (14.67%) | N = 10,748 (14.64%) | N = 51,910 (70.70%) | |
| Any Adverse Event | 1,693 (15.72%) | 1,697 (15.79%) | 7,998 (15.41%) | 0.491 |
| Death | 682 (6.33%) | 686 (6.38%) | 3,464 (6.67%) | 0.289 |
| Myocardial Infarction | 107 (0.99%) | 99 (0.92%) | 505 (0.97%) | 0.847 |
| Stroke | 69 (0.64%) | 79 (0.74%) | 336 (0.65%) | 0.573 |
| **Renal Failure** | **180 (1.67%)** | **135 (1.26%)** | **595 (1.15%)** | **<0.001** |
| **Pulmonary Embolism** | **95 (0.88%)** | **62 (0.58%)** | **329 (0.63%)** | **0.008** |
| Congestive Heart Failure | 106 (0.98%) | 122 (1.14%) | 550 (1.06%) | 0.558 |
| **Pneumonia** | **315 (2.93%)** | **266 (2.47%)** | **1,214 (2.34%)** | **0.002** |

**Table 3. Incidence of surgical adverse events following all hip fracture surgeries, organized by fellowship type of the treating physician.**

| Type: Hip Fracture | Trauma | Adult Reconstruction | Other | Univariate P-value |
|---|---|---|---|---|
| N = 73,427 (100%) | N = 10,769 (14.67%) | N = 10,748 (14.64%) | N = 51,910 (70.70%) | |
| **Any Surgical Complication** | **586 (5.44%)** | **639 (5.95%)** | **3,215 (6.19%)** | **0.011** |
| Bone Fracture | 179 (1.66%) | 164 (1.53%) | 776 (1.49%) | 0.435 |
| **Dislocation** | **47 (0.44%)** | **93 (0.87%)** | **261 (0.50%)** | **<0.001** |
| Infection | 141 (1.31%) | 115 (1.07%) | 651 (1.25%) | 0.218 |
| Nonunion/Delayed Union | 42 (0.39%) | 64 (0.60%) | 277 (0.53%) | 0.088 |
| **Skin Ulcer/Blister** | **40 (0.37%)** | **68 (0.63%)** | **324 (0.62%)** | **0.006** |
| Implant Failure | 147 (1.37%) | 129 (1.20%) | 723 (1.39%) | 0.292 |
| Nerve Palsy/Injury | 20 (0.19%) | 29 (0.27%) | 125 (0.24%) | 0.423 |
| **Hemorrhage** | **16 (0.15%)** | **20 (0.19%)** | **235 (0.45%)** | **<0.001** |
| Limb Ischemia | 1 (0.01%) | 4 (0.04%) | 7 (0.01%) | 0.178 |
| Tendon Ligament/Injury | 2 (0.02%) | 0 (0.00%) | 9 (0.02%) | 0.388 |
| Compartment Syndrome | 0 (0.00%) | 0 (0.00%) | 1 (0.00%) | 0.813 |
| Wrong Side/Site Surgery | 1 (0.01%) | 0 (0.00%) | 3 (0.01%) | 0.642 |
| Vascular Injury | 3 (0.03%) | 3 (0.03%) | 10 (0.02%) | 0.771 |

**Table 4. Incidence of surgical adverse events by fracture pattern, organized by fellowship type of the treating physician.**

| Type: Hip Fracture | Trauma | Adult Reconstruction | Other | Univariate P-value |
|---|---|---|---|---|
| N = 73,427 (100%) | N = 10,769 (14.67%) | N = 10,748 (14.64%) | N = 51,910 (70.70%) | |
| Simple Percutaneous Pinning Hip Fracture (27235) | 53 (4.71%) | 46 (4.26%) | 311 (5.34%) | 0.270 |
| Femoral Neck Hip Fracture (27236) | 266 (7.36%) | 311 (7.72%) | 1,443 (7.86%) | 0.590 |
| **Intertrochanteric Hip Fracture (27244, 27245)** | **267 (4.43%)** | **282 (5.00%)** | **1,461 (5.27%)** | **0.025** |

specifically attributed to dislocation (p<0.001), skin ulcer/blister (p = 0.006), and hemorrhage (p < 0.001). There were no significant differences detected in the rates of bone fracture, infection, nonunion/delayed union, implant failure, nerve palsy/injury, lib ischemia, tendon ligament/injury, compartment syndrome, wrong side/site surgery, or vascular injury. When compared by fracture pattern, trauma surgeons had a slightly statistically significant reduction in surgical adverse events when performing procedures on intertrochanteric hip fractures compared to their counterparts from adult reconstruction or other fellowship categories (Table 4).

**Table 5. Incidence of surgical adverse events following all hip fracture surgeries, organized by case volume of the treating physician.**

| Type: Hip Fracture | 0–15 Cases | 16–30 Cases | 31+ Cases | Univariate P-value |
|---|---|---|---|---|
| N = 73,427 (100%) | N = 36,679 (49.95%) | N = 26,792 (36.49%) | N = 9,956 (13.56%) | |
| **Any Surgical Complication** | **2,377 (6.48%)** | **1,605 (5.99%)** | **458 (4.60%)** | **<0.001** |
| **Bone Fracture** | **602 (1.64%)** | **407 (1.52%)** | **110 (1.10%)** | **0.001** |
| Dislocation | 183 (0.50%) | 168 (0.63%) | 50 (0.50%) | 0.079 |
| Infection | 459 (1.25%) | 327 (1.22%) | 121 (1.22%) | 0.924 |
| **Nonunion/Delayed Union** | **214 (0.58%)** | **128 (0.48%)** | **41 (0.41%)** | **0.050** |
| **Skin Ulcer/Blister** | **238 (0.65%)** | **156 (0.58%)** | **38 (0.38%)** | **0.008** |
| **Implant Failure** | **582 (1.59%)** | **320 (1.19%)** | **97 (0.97%)** | **<0.001** |
| Nerve Palsy/Injury | 92 (0.25%) | 68 (0.25%) | 14 (0.14%) | 0.104 |
| **Hemorrhage** | **125 (0.34%)** | **133 (0.50%)** | **13 (0.13%)** | **<0.001** |
| Limb Ischemia | 6 (0.02%) | 6 (0.02%) | 0 (0.00%) | 0.328 |
| Tendon Ligament/Injury | 7 (0.02%) | 2 (0.01%) | 2 (0.02%) | 0.450 |
| Compartment Syndrome | 1 (0.00%) | 0 (0.00%) | 0 (0.00%) | 0.606 |
| Wrong Side/Site Surgery | 3 (0.01%) | 0 (0.00%) | 1 (0.01%) | 0.309 |
| Vascular Injury | 6 (0.02%) | 6 (0.02%) | 4 (0.04%) | 0.359 |

When reviewing surgeon-reported surgical adverse events by the treating physician's case volume, increasing case volume, comparing 0–15 cases, 16–30 cases, and 31+ cases, significantly reduced surgical complications (p<0.001) (Table 5). In particular, surgical complications including bone fracture (p = 0.001), skin ulcer/blister (p = 0.008), implant failure (p<0.001), and hemorrhage (p<0.001) were significantly reduced with increased case volumes and nonunion/delayed union (p = 0.050) less significantly. The implications of case volume was further evident in the incidence of surgical adverse events in femoral neck hip fracture and intertrochanteric hip fracture procedures (p<0.001) (Table 6).

Multivariate regression revealed four factors independently associated with reduced complication rates and three factors associated with increased complication rates. Patient age (OR 0.91; 95% CI: 0.88–0.95; p<0.001), male gender (OR 0.91; 95% CI: 0.85–0.97; p = 0.006), and case volume of either 16–30 cases (OR 0.91; 95% CI: 0.85–0.97; p = 0.003) or 31+ cases (OR 0.68; 95% CI: 0.61–0.76; p<0.001) were found to be protective factors against complications (Table 7). Patients treated in the Midwest (OR 1.21; 95% CI: 1.10–1.33; p<0.001) or South (OR 1.11; 95% CI: 1.00–1.22; p = 0.045) region of the United States were more likely to experience a complication than those treated in other regions. Finally, patients treated for a femoral neck hip fractures (OR 1.61; 95% CI: 1.44–1.80; p<0.001) were more likely to experience complications. Fellowship status did not appear to affect complication rates.

## Discussion

In recent years, the number of orthopaedic residency graduates seeking fellowship training increased significantly [23, 28]. In fact, orthopaedics has one of the highest fellowship

**Table 6. Incidence of surgical adverse events by fracture pattern, organized by case volume of the treating physician.**

| Type: Hip Fracture | 0–15 Cases | 16–30 Cases | 31+ Cases | Univariate P-value |
|---|---|---|---|---|
| N = 73,427 (100%) | N = 36,679 (49.95%) | N = 26,792 (36.49%) | N = 9,956 (13.56%) | |
| Simple Percutaneous Pinning Hip Fracture (27235) | 235 (5.48%) | 132 (4.84%) | 43 (4.28%) | 0.220 |
| **Femoral Neck Hip Fracture (27236)** | **1,001 (8.14%)** | **785 (7.87%)** | **234 (6.29%)** | **<0.001** |
| **Intertrochanteric Hip Fracture (27244, 27245)** | **1,141 (5.68%)** | **688 (4.88%)** | **181 (3.46%)** | **<0.001** |

**Table 7. Factors independently associated with surgical complications among all hip fracture surgeries.**

| Factor | Likelihood of Surgical Complication | | |
|---|---|---|---|
| N = 73,427 (100%) | OR | 95% CI | P-value |
| **Patient Age (per decade)** | **0.91** | **[0.88–0.95]** | **<0.001** |
| **Patient Sex** | | | |
| Female | 1.00 | – | – |
| **Male** | **0.91** | **[0.85–0.97]** | **0.006** |
| **Region** | | | |
| Northeast | 1.00 | – | – |
| Northwest | 1.10 | [0.96–1.27] | 0.185 |
| **Midwest** | **1.21** | **[1.10–1.33]** | **<0.001** |
| **South** | **1.11** | **[1.00–1.22]** | **0.045** |
| Southeast | 1.01 | [0.91–1.12] | 0.848 |
| Southwest | 1.03 | [0.93–1.14] | 0.609 |
| **Case Volume of Treating Physician** | | | |
| 0–15 cases | 1.00 | – | – |
| **16–30 cases** | **0.91** | **[0.85–0.97]** | **0.003** |
| **31+ cases** | **0.68** | **[0.61–0.76]** | **<0.001** |
| **Fellowship Status** | | | |
| Other | 1.00 | – | – |
| Trauma | 0.97 | [0.86–1.09] | 0.613 |
| Adult Reconstruction | 0.99 | [0.90–1.09] | 0.827 |
| **Procedure Type** | | | |
| Simple Percutaneous Pinning Hip Fracture (27235) | 1.00 | – | – |
| **Femoral Neck Hip Fracture (27236)** | **1.61** | **[1.44–1.80]** | **<0.001** |
| Intertrochanteric Hip Fracture (27244, 27245) | 1.02 | [0.92–1.14] | 0.692 |

participation rates across specialties–as high as 90% by some estimates [7, 28, 29]. Furthermore, physicians are more likely to perform procedures within their given specialty thus making patients with any given pathology more likely to be treated by a surgeon who was fellowship-trained in the most relevant specialty [23]. Given that fellowship training requires additional time and significant financial resources, it is becoming increasingly important to understand the effects of fellowship-training on surgeons' clinical practice and the value for decreasing patient complications.

In this series of 73,427 hip fractures, patients treated by surgeons who were fellowship-trained in orthopaedic trauma were more likely to experience renal failure, pulmonary emboli, and pneumonia in the post-operative course than those treated by other specialties. We suspect this may be a reflection of traumatologists treating more acute and perhaps medically complex patients than other subspecialties, but in the absence of patient comorbidity data in the ABOS dataset, further information is required to support or negate this hypothesis.

Initial analysis also revealed that the trauma-trained physicians also had fewer overall surgical complications than the other cohorts. However, only three specific complications occurred at statistically different rates amongst the three groups. Dislocations, skin ulcers/blisters, and hemorrhage all occurred at a significantly lower frequencies in the trauma cohort than either the adult reconstruction or other categories. Subgroup analysis by fracture pattern revealed that only intertrochanteric hip fractures had differing rates of complication based on fellowship training, with trauma fellowship-trained surgeons experiencing the lowest complication rates.

Prior studies investigating fellowship status for hip fracture patients have shown mixed results. In a retrospective study of 298 femoral neck fractures, Mabry et al. found that adult reconstruction trained surgeons had shorter operative duration and less complications than those treated by general practice orthopaedic surgeons when performing hip hemiarthroplasty [22]. Decreased complications were also shown in a separate study for total hip arthroplasty [30]. Patients treated by trauma trained doctors had longer delays to surgery, possibly reflecting scheduling conflicts and adversely affecting complication incidence. Additionally, patients treated by traumatologists were significantly more likely to have suffered prior myocardial infarctions or be diagnosed with chronic obstructive pulmonary disease, presenting potential confounders [22]. In a conflicting retrospective study of 871 patients by Yuan and Kwek, they found reduced surgical delays and shorter surgical times for trauma surgeons when compared to non-trauma surgeons but found no difference in the incidence of postoperative complications and mortality [5]. However, both of these studies are single center studies, which may have limited external validity, as the skill sets and time restraints of individual surgeons at these respective centers may have influenced results.

The absence of effect of fellowship training on hip fracture surgeon-reported complications when compared to procedures such as joint arthroplasty may be attributable to the common practice of hip fracture call regardless of surgical subspecialty, which likely is a fundamental skill acquired during orthopaedic surgery residency training. While elective procedures like THA may largely be concentrated to adult reconstruction fellowship-trained surgeons in early practice, general orthopaedic or hip fracture call is broadly dispersed thus enabling surgeons from any subspecialty to gain considerable caseloads. This would not be true in reverse, thus muddling conclusions about value of fellowship training for hip fracture complications. Additionally, this current study uses data from ABOS candidates who are in their initial stages of their career and this finding may differ after years of being an attending physician.

Although fellowship status did not seem to significantly influence complication rates in the current study, surgeons' annual hip fracture case volume did seem to influence the rates. Across the orthopedic literature, the relationship between surgeon and hospital volume and complications rates has been shown in the settings of hip or knee arthroplasty as well as for the operative treatment of scoliosis [31, 32]. In our study, surgeons treating more than 30 hip fractures during the collection period demonstrated an odds ratio of 0.68 in comparison to surgeons treating 15 fractures or less per year. This finding is similar to findings in studies about total knee arthroplasty (TKA) in which surgeons performing higher numbers of TKA's reported fewer early complications than surgeons with less experience [33].

Our findings regarding case volume for hip fractures are in contrast to those by Okike et al., who used hip fracture registry data to assess the relationship between hospital or surgeon volume against patient morbidity and mortality [20]. In that study, there was no association between case volume and reoperation, medical complications, or readmissions, however, despite it being a registry study, the data is limited in number and to one geographic region. The discrepancy between these findings and those of the presently presented data warrants further exploration of the relationship between surgeon volume and complication rates.

Our findings also differ from those of Spaans et al. [21]. In their retrospective cohort from the Netherlands, patients who were treated for femoral neck fractures with hemiarthroplasty by low-volume surgeons, defined as less than 10 arthroplasties per year, were compared to those treated by moderate-volume (10–35 arthroplasties per year) and high-volume (35 + cases/year). The authors found that neither case volume nor surgeon experience correlated with hip prosthesis survival, patient mortality, surgical site infection, periprosthetic fracture, or prosthesis dislocation, however, this study too was limited in patient numbers (752 cases) and to one academic hospital.

Finally, in a recent systematic review by Wiegers et al., their findings were mixed with some findings supporting studies such as Okike and Spaans with no effect observed between hospital or surgeon volume on complications in the setting of hip fractures [34]. These associations were limited to morbidity and infectious complications, however. This finding does fall in line with some aspects of our results, in which infection was not found to be statistically different. Conversely, they did find a relationship between high surgeon volume and shorter lengths of stay [18, 35, 36]. As shorter lengths of stay after hip fractures have been shown to reduce rates of early mortality, this may also support the case volume findings of the current study [37, 38].

The presented project has several limitations. The inherent biases of any retrospective review hold true in this study. Moreover, the ABOS dataset does not provide granular patient comorbidity data thus limiting sub-group analysis conclusions. Additionally, because it is surgeon self-reported data there may be errors in inputting, missed cases, and/or complications may have been classified different depending on the orthopedist group; however, the ABOS database is considered to be highly reliable and extremely robust as the six months of case collection data is submitted as a mandatory part of the board certification for new orthopaedic surgeons, thus cases are inspected carefully and self-reported data is monitored [24]. Thus, the current study offers important insights into refuting the conclusions of previous studies wherein surgeon case volume did not affect complication rates and highlights the need for future research in this domain.

## Conclusion

Based on our findings, fellowship status of training orthopaedic surgeons showed no major difference in the overall incidence of surgeon-reported surgical complications for hip fracture patients, and likely demonstrates the fundamental competence of orthopaedic surgeons in managing hip fractures following the successful completion of orthopaedic residency training. Despite such minor differences attributed to the fellowship type, case volume of the performing surgeon proved to have a more significant association with decreased risk of surgeon-reported surgical complications. The importance of case volume showcases the importance of additional training, thus potentially identifying the importance of concentrating hip fracture call in an early surgeon's career. Future studies should review surgeon data later in their career to analyze these impact of early training and continued case volume as it pertains to more veteran surgeons and their medical and surgical complication rates in hip fracture procedures. Additional studies may want to further investigate the role of patient factors, including comorbidities and case acuity, as these may play a role in relation to surgeon factors, affecting the complexity and technical nature of the procedure.

## Author Contributions

**Conceptualization:** Jonathan N. Grauer, Lee E. Rubin.

**Data curation:** Taylor D. Ottesen, Michael R. Mercier, Jordan Brand, Jonathan N. Grauer, Lee E. Rubin.

**Formal analysis:** Taylor D. Ottesen, Michael R. Mercier, Jordan Brand, Michael Amick, Jonathan N. Grauer, Lee E. Rubin.

**Investigation:** Taylor D. Ottesen, Michael R. Mercier, Jordan Brand, Jonathan N. Grauer, Lee E. Rubin.

**Methodology:** Taylor D. Ottesen, Michael R. Mercier, Jonathan N. Grauer, Lee E. Rubin.

**Writing – original draft:** Taylor D. Ottesen, Jordan Brand, Michael Amick, Jonathan N. Grauer, Lee E. Rubin.

**Writing – review & editing:** Taylor D. Ottesen, Michael R. Mercier, Jordan Brand, Michael Amick, Jonathan N. Grauer, Lee E. Rubin.

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
