## [Decision Letter · Decision Letter 0]

8 Oct 2021

PONE-D-21-28043The Case for Decreased Surgeon-reported Complications Due to Surgical Volume and Fellowship Status in the Treatment of Geriatric Hip Fracture: An Analysis of the ABOS DatabasePLOS ONE

Dear Dr. Grauer,

Thank you for submitting your manuscript to PLOS ONE. After careful consideration, we feel that it has merit but does not fully meet PLOS ONE’s publication criteria as it currently stands. Therefore, we invite you to submit a revised version of the manuscript that addresses the points raised during the review process.

Both reviewers made ever effort to give comments and recomendations to improve your manuscript.Please, follow the remarks (especially of reviewers 1) and include his ideas in the revised manuscript.

We look forward to receiving your revised manuscript.

Kind regards,

Hans-Peter Simmen, M.D., Professor of Surgery

Academic Editor

PLOS ONE

Journal Requirements:

Reviewers' comments:

Reviewer's Responses to Questions

**Comments to the Author**

1. Is the manuscript technically sound, and do the data support the conclusions?

Reviewer #1: Yes

Reviewer #2: Yes

2. Has the statistical analysis been performed appropriately and rigorously? 

Reviewer #1: I Don't Know

Reviewer #2: Yes

3. Have the authors made all data underlying the findings in their manuscript fully available?

Reviewer #1: No

Reviewer #2: No

4. Is the manuscript presented in an intelligible fashion and written in standard English?

Reviewer #1: Yes

Reviewer #2: Yes

5. Review Comments to the Author

Reviewer #1: Dear Editor,

It was a pleasure for me to review the above-mentioned manuscript for PLOS-ONE.

AIM OF THE STUDY

The aim of this study is important as the final goal of any fellowship training (providing better care to a defined sub-population of orthopaedic patients) still needs to be scientifically investigated, at least for trauma patients. Right now, fellowship is recognized as the “right to operate on some patients” by academics, hospital/clinic administrators and sometimes even by patients, but this “right” relies on few objective data.

LEVEL OF ENGLISH AND EDITION

Minimal corrections needed. See one of my comments.

QUALITY OF THE STUDY/MANUSCRIPT

-General:

On the Plos One Editorial Manager page, corresponding author is Jonathan N Grauer, whereas on the title page, corresponding author is Taylor D Ottesen. Please correct one of these statements to be consistent throughout the manuscript.

-ABSTRACT:

-Lines 47-48: “no prior study has investigated…”; however, in the Introduction, the authors cite references 5 & 23 (lines 114-117). Please modify the abstract, stating something like “Rare studies…” or “Scarce studies…” or something else.

-Line 61: after reading the Methods and the Results chapters, “1994-2004” should be replaced by “1999-2004”. Also see my comments on percentages (9.43% -> 6.8%, 60.92% -> 22%) on Results/Table 1.

-MANUSCRIPT

-Introduction:

-Line 95: replace “including” by “includes”.

-Lines 120-121: the statement should be mitigated, as references 5 & 23 studied this point.

-Materials and methods:

-Line 149: define the abbreviation “CPT” at its first appearance (line 149), not at its second appearance (Line 158).

-“Data” section: please define which states are included in each of the 6 regions. This might be obvious for US (and maybe Canadian) readers, but not for the rest of the world. As a non-US reviewer, I am for instance interested in incongruous/off-the-grid questions, such as: to which region belong Alaska and Hawaii? Were data from Puerto Rico, US Virgin Islands and other overseas territories included? This could be described in a Figure or a Table to save word count.

-“Data” section: “Number of cases completed by the candidate during…”. Does this number really reflect the volume? Might underestimation be possible? Please add a short statement about this issue, either in the M&M chapter, or in the limitation section.

Results:

-I am a little bit surprised by the fact that the tiny differences in ages (82.2 +/- 8.2 vs. 82.8 +/- 7.8 vs. 82.6 +/- 7.8, Line 199) and gender repartition (71,6% female vs. 73.2% vs. 73.5%, Line 201) among different fellowship groups reaches statistical significance which such mighty p-values. I would simply ask the authors to double-check these results (this also applies to Table 1).

-Lines 201-204/Table 1:

1° see my comment on abstract Line 61.

2° I think percentages in Table 1 should be corrected. For example: In years 1999-2004, 14954 procedures were performed (1016 + 1894 + 12044), meaning that Trauma fellowship trained surgeons performed 6.8% of the procedures (1016 out of 14954, not 1016 out of 10796), Adult reco 12.7% of the procedures (1894 out of 14954) and others 80.5% (12044 out of 14954). For years 2005-2010, my calculations result in 11.1%, 14% and 74.9% respectively. For years 2011-2016, 22%, 16.3% and 61.7% respectively; P-value should thus be recalculated, but the interpretative text is probably still ok.

-Table 1:

1° I think after calculation that N for “Other” should be 51910 rather than 60612. However, percentages for the whole column seem to be generally correct:

Total N for Sex/Other, Year/Other and Procedure type/Other is 51910-ok;

Total N for Region/Other is 51643-please correct or add a line for “n.a. region” or “other region”.

P-values might have to be corrected.

2° Total N for Region/Trauma is 10710 and does not correspond to 10769: please correct or add a line for “n.a. region” or “other region”.

3° Total N for Region/Adult reco is 10732 and does not correspond to 10748: please correct or add a line for “n.a. region” or “other region”.

-Table 2: I think after calculation that N for other should be 51910 rather than 60612. However, percentages for the whole column seem to be correct. P-values might have to be corrected.

-Tables 3a and 3b: I think after calculation that N for other should be 51910 rather than 60612. However, percentages for the whole column seem to be correct. P-values might have to be corrected.

Discussion:

-As long as corrections following my remarks on the Results section and Tables do not provoke unexpected changes in the interpretation of results, this section won’t need any corrections to be performed.

Reviewer #2: Dear Editor,

Dear Authors,

Thank you very much for the opportunity to review this interesting manuscript. Authors could show decreased surgeon reported complications due to surgical volume. This is a nicely written paper. I would like to suggest to address the following issues in order to improve the quality of the manuscript.

Introduction:

- Aims of the study were described.

Methods:

- Did all patients signed the informed consent?

- Do you need to have ethical approval to use this data?

- Authors should note that the statistical analysis in big data is not always useful. Beside the regression analysis, other statistics has mainly a descriptive character.

Results

- What about the time point of surgery? Is it known (< 24 h) or later?

- The regression analysis is the main part of results.

Discussion

- Nice Discussion and description of limitations

6. PLOS authors have the option to publish the peer review history of their article (what does this mean?). If published, this will include your full peer review and any attached files.

Reviewer #1: **Yes: **Axel GAMULIN

Reviewer #2: **Yes: **Roman Pfeifer

---

## [Author Response · Author response to Decision Letter 0]

10 Jan 2022

Dear Editorial Board,

Attached is a copy of our revised manuscript entitled “The Case for Decreased Surgeon-reported Complications Due to Surgical Volume and Fellowship Status in the Treatment of Geriatric Hip Fracture: An Analysis of the ABOS Database”. We wish to thank you and the reviewers for their time and effort in their critical review of our manuscript. We have responded to the requested changes as detailed below and believe the reviewers’ comments have allowed us to substantially improve the quality of the manuscript. Our responses are recorded in italics with changes to the manuscript documented in bold. 

Warmly,

Taylor D. Ottesen, M.D., M.B.A

Jonathan N. Grauer, MD

Lee E. Rubin, M.D. FAAOS, FAAHKS, FAOA

Response to Reviewers:

Reviewer #1:

-General:

On the Plos One Editorial Manager page, corresponding author is Jonathan N Grauer, whereas on the title page, corresponding author is Taylor D Ottesen. Please correct one of these statements to be consistent throughout the manuscript.

Thank you for identifying this discrepancy. We have corrected the cover letter submitted with the most recent version of the manuscript to identify Dr. Jonathan N Grauer as the corresponding author. The title page and PLOS One Editorial Manager page are now in agreement with the designation.

 Corresponding Author:

Jonathan N. Grauer, M.D.

Professor of Orthopaedics & Rehabilitation; Associate Dean, Faculty Affairs; Director of Orthopaedic Spine Service, Orthopaedics & Rehabilitation

Department of Orthopaedics and Rehabilitation, Yale School of Medicine

800 Howard Avenue, New Haven, CT 06519, USA

Phone: 203-228-2622 

Email: Jonathan.grauer@yale.edu

Abstract: 

-Lines 47-48: “no prior study has investigated…”; however, in the Introduction, the authors cite references 5 & 23 (lines 114-117). Please modify the abstract, stating something like “Rare studies…” or “Scarce studies…” or something else.

Thank you for this comment. We agree that clarifying the existence of scarce studies as opposed to the previously stated no studies better reports the current state of the literature. To correct this statement, we have reframed the sentence to identify the scarcity of studies that exist in the literature and the importance of further examination of the subject. Below, we have included the modified sentence as it now reads. We thank the reviewer for bringing this to our attention.

One proposed benefit of fellowship training is decrease in complications, however, scarce studies has investigated the rates of medical and surgical complications for hip fracture patients between orthopedists from different fellowship backgrounds.

-Line 61: after reading the Methods and the Results chapters, “1994-2004” should be replaced by “1999-2004”. Also see my comments on percentages (9.43% -> 6.8%, 60.92% -> 22%) on Results/Table 1.

Thank you to the reviewer for their careful review. We agree the year included (1994) was an oversight in this line of text and should correctly read 1999. We have modified this year, and we have included the corrected sentence below for reference. We thank the reviewer for noting these important details.

An increasing number of hip fractures are being treated by trauma fellowship trained surgeons (9.43% in 1999-2004 to 60.92% in 2011-2016).

Manuscript

-Introduction:

-Line 95: replace “including” by “includes”.

-Lines 120-121: the statement should be mitigated, as references 5 & 23 studied this point.

Thank you to the reviewer for these comments. We agree the word “including” should be replaced by the word “includes”. We have made the respective change and provided the updated sentence as it reads below. Additionally, we agree with the point regarding the nature of the current literature on the relationship between fellowship training and post-operative complications. To add clarity, we have reframed the sentence to acknowledge the presence of research but more importantly highlight the insufficient evaluation that exists thus far. This paper and further papers are required to further understand the role of fellowship in the training of orthopaedic surgeons as it pertains to outcomes and complications. With this rationale, we have opted for the word “understudied” to highlight this point with the emphasis on this paper and future papers to continue to broaden the literature on this topic. We have included the updated sentence as it reads in the text now. We thank the reviewer for their comments.

Treatment for hip fractures varies from hospital to hospital, and often includes treatment by general orthopaedic surgeons, orthopaedic trauma surgeons, or other fellowship trained orthopaedic surgeons

Additionally, the act of fellowship training itself on post-operative complications remains understudied.

Materials and methods: 

-Line 149: define the abbreviation “CPT” at its first appearance (line 149), not at its second appearance (Line 158).

Thank you to the reviewer for this comment. We agree with the importance of identifying abbreviations at their first appearance. We have modified the sentences accordingly to define the abbreviation in full at its first appearance and the removal of the definition in its previous location. We have provided the updated sentence as it now reads with the definition provided at its first appearance. We thank the reviewer for catching these details.

The list of procedures is logged by the applicant and includes a wide range of data including the applicant’s fellowship status, patient demographics (e.g. patient age and gender) and information about the procedure (e.g. Current Procedural Terminology (CPT) codes, and anesthetic, medical, and surgical complications).

-“Data” section: please define which states are included in each of the 6 regions. This might be obvious for US (and maybe Canadian) readers, but not for the rest of the world. As a non-US reviewer, I am for instance interested in incongruous/off-the-grid questions, such as: to which region belong Alaska and Hawaii? Were data from Puerto Rico, US Virgin Islands and other overseas territories included? This could be described in a Figure or a Table to save word count.

Thank you for this suggestion. We agree that this is helpful for a global audience. Below are the different regions and the associated states which we have put into a figure form and included in the text in line 181 and included as Figure 1:

Northeast - Connecticut, Massachusetts, Maine, New Hampshire, Rhode Island, Vermont, New Jersey, New York, Puerto Rico, Delaware, Maryland, Pennsylvania, DC

Southeast - Virginia, West Virginia, North Carolina, South Carolina, Florida, Georgia

South - Alabama, Kentucky, Mississippi, Tennessee, Arkansas, Louisiana, Texas

Southwest - New Mexico, Oklahoma, Arizona, California, Hawaii, Nevada, Colorado

Northwest - Alaska, Idaho, Oregon, Washington, Montana, North Dakota, South Dakota, Utah, Wyoming

Midwest - Iowa, Kansas, Nebraska, Missouri, Illinois, Indiana, Michigan, Minnesota, Ohio, Wisconsin

-“Data” section: “Number of cases completed by the candidate during…”. Does this number really reflect the volume? Might underestimation be possible? Please add a short statement about this issue, either in the M&M chapter, or in the limitation section.

Yes, this number should truly reflect volume. ABOS candidates are required to enter every surgical case from every hospital/surgery center in which they operate during the entire case collection period. If they fail to do this, they may be ineligible for certification. Further, hospitals submit their logs of all cases done and these must match. However, as the reviewer points out, there is always a possibility that case documentation could be missed. We have added this to the limitations section of the manuscript.

Results:

-I am a little bit surprised by the fact that the tiny differences in ages (82.2 +/- 8.2 vs. 82.8 +/- 7.8 vs. 82.6 +/- 7.8, Line 199) and gender repartition (71,6% female vs. 73.2% vs. 73.5%, Line 201) among different fellowship groups reaches statistical significance which such mighty p-values. I would simply ask the authors to double-check these results (this also applies to Table 1).

We sincerely thank this reviewer for their astute review of the manuscript’s tables and figures. The mean age and the standard deviation for each fellowship subgroup was re-checked, in addition to the accompanying ANOVA test p-value used to assess statistical significance. No difference was noted. This finding appears to be statistically significant likely due to the sheer size of the database utilized. However, we acknowledge that this finding is difficult to interpret and does not appear to have any meaningful clinical significance.

-Lines 201-204/Table 1: 

1° see my comment on abstract Line 61.

This was corrected in the abstract per your suggestion. Thank you!

2° I think percentages in Table 1 should be corrected. For example: In years 1999-2004, 14954 procedures were performed (1016 + 1894 + 12044), meaning that Trauma fellowship trained surgeons performed 6.8% of the procedures (1016 out of 14954, not 1016 out of 10796), Adult reco 12.7% of the procedures (1894 out of 14954) and others 80.5% (12044 out of 14954). For years 2005-2010, my calculations result in 11.1%, 14% and 74.9% respectively. For years 2011-2016, 22%, 16.3% and 61.7% respectively; P-value should thus be recalculated, but the interpretative text is probably still ok.

Thank you for the thoughtful approach of how percentages should be calculated in Table 1 for the “year of procedure”. The decision to calculate percentages vertically (within each fellowship type), or horizontally (within each 5-year time period) was a difficult one that our research team debated for quite some time. We agree with your assessment, and have recalculated percentages and p-values accordingly and put a note at the bottom of the table. Notably, the P-value for this chi-square analysis remained significant (p<0.001). The interpretative text of this trend analysis will thus remain the same.

-Table 1: 

1° I think after calculation that N for “Other” should be 51910 rather than 60612. However, percentages for the whole column seem to be generally correct:

 Total N for Sex/Other, Year/Other and Procedure type/Other is 51910-ok; 

Total N for Region/Other is 51643-please correct or add a line for “n.a. region” or “other region”.

P-values might have to be corrected. 

2° Total N for Region/Trauma is 10710 and does not correspond to 10769: please correct or add a line for “n.a. region” or “other region”. 

3° Total N for Region/Adult reco is 10732 and does not correspond to 10748: please correct or add a line for “n.a. region” or “other region”.

Thank you for the astute observations here. The N for the “Other” column has been adjusted, and the percentages have remained the same. A line for “Other” has been added to the region analysis. Notably, P-values have been recalculated, but remain the same in all instances (P<0.001), given the small amount of missing data.

-Table 2: I think after calculation that N for other should be 51910 rather than 60612. However, percentages for the whole column seem to be correct. P-values might have to be corrected. 

Thank you once again for the astute observations. The N for the “Other” column has once again been adjusted, and the percentages have remained the same. The percentages and accompanying p-values were correct as initially stated and thus have not changed. 

-Tables 3a and 3b: I think after calculation that N for other should be 51910 rather than 60612. However, percentages for the whole column seem to be correct. P-values might have to be corrected. 

The same suggested changes for table 2 have been made for table 3 as well. Thank you!

Discussion:

-As long as corrections following my remarks on the Results section and Tables do not provoke unexpected changes in the interpretation of results, this section won’t need any corrections to be performed.

Given that the percentages and accompanying p-values were correct as initially stated and thus have not changed, the interpretation of the results has not changed either and has been left the same. Thank you for your help to make this manuscript the best it can be and pointing out important and necessary changes and edits. We appreciate it!

Reviewer #2: 

Dear Authors,

Thank you very much for the opportunity to review this interesting manuscript. Authors could show decreased surgeon reported complications due to surgical volume. This is a nicely written paper. I would like to suggest to address the following issues in order to improve the quality of the manuscript.

Thank you so much for reviewing our manuscript and the resulting suggestions. We appreciate the reviewer taking the time to help make our manuscript the best it can be.

Introduction:

- Aims of the study were described.

Thank you!

Methods:

- Did all patients signed the informed consent?

The current study is using the American Board of Orthopaedic Surgeons (ABOS) dataset which is blinded data from across the country. All data entered into this database is done by ABOS examinees for which patients had to sign consent for the surgery and utilization of these details for reporting purposes.

- Do you need to have ethical approval to use this data?

Yes. The ABOS Research Committee approved the current study and the study was classified as exempt and received ethical approval by our institutional review board.

- Authors should note that the statistical analysis in big data is not always useful. Beside the regression analysis, other statistics has mainly a descriptive character.

Thank you for the comment. We agree that big data is not always useful, however, the current study tried to use the most appropriate statistical tests for the situation and to control for all available factors to try to give the most accurate interpretation of the data. We feel the findings support important training trends and the need for specialty specific fellowship exposure.

Results

- What about the time point of surgery? Is it known (< 24 h) or later?

Unfortunately this is a limitation of the dataset. It is not recorded at what time point the patient received surgery following the injury. This is a good question for future studies.

- The regression analysis is the main part of results.

Thank you for this comment. Due to the limited nature of the dataset, this analysis was the most complete statistical test to explore associations between data points.

Discussion

- Nice Discussion and description of limitations

Thank you for the feedback! We appreciate your support.

---

## [Editor Report · Decision Letter 1]

20 Jan 2022

The Case for Decreased Surgeon-reported Complications Due to Surgical Volume and Fellowship Status in the Treatment of Geriatric Hip Fracture: An Analysis of the ABOS Database

PONE-D-21-28043R1

Dear Dr. Grauer,

We’re pleased to inform you that your manuscript has been judged scientifically suitable for publication and will be formally accepted for publication once it meets all outstanding technical requirements.

Kind regards,

Hans-Peter Simmen, M.D., Professor of Surgery

Academic Editor

PLOS ONE
---

## [Editor Report · Acceptance letter]

14 Feb 2022

PONE-D-21-28043R1 

The Case for Decreased Surgeon-reported Complications Due to Surgical Volume and Fellowship Status in the Treatment of Geriatric Hip Fracture: An Analysis of the ABOS Database 

Dear Dr. Grauer:

I'm pleased to inform you that your manuscript has been deemed suitable for publication in PLOS ONE. Congratulations! Your manuscript is now with our production department. 

Kind regards, 

on behalf of

Dr. Hans-Peter Simmen 

Academic Editor

PLOS ONE